# Efficient Generation and Correction of Mutations in Human iPS Cells Utilizing mRNAs of CRISPR Base Editors and Prime Editors

**DOI:** 10.3390/genes11050511

**Published:** 2020-05-06

**Authors:** Duran Sürün, Aksana Schneider, Jovan Mircetic, Katrin Neumann, Felix Lansing, Maciej Paszkowski-Rogacz, Vanessa Hänchen, Min Ae Lee-Kirsch, Frank Buchholz

**Affiliations:** 1Medical Systems Biology, Medical Faculty and University Hospital Carl Gustav Carus, TU Dresden, 01307 Dresden, Germany; duran.sueruen@tu-dresden.de (D.S.); aksana.schneider@tu-dresden.de (A.S.); jovan.mircetic@tu-dresden.de (J.M.); felix.lansing@tu-dresden.de (F.L.); maciej.paszkowski-rogacz@tu-dresden.de (M.P.-R.); 2Mildred Scheel Early Career Center, National Center for Tumor Diseases Dresden (NCT/UCC), Medical Faculty and University Hospital Carl Gustav Carus, TU Dresden, 01307 Dresden, Germany; 3Stem Cell Engineering Facility, Center for Molecular and Cellular Bioengineering (CMCB), TU Dresden, 01307 Dresden, Germany; katrin.neumann1@tu-dresden.de; 4Department of Pediatrics, Medical Faculty and University Hospital Carl Gustav Carus, TU Dresden, 01307 Dresden, Germany; Vanessa.Haenchen@uniklinikum-dresden.de (V.H.); minae.lee-kirsch@uniklinikum-dresden.de (M.A.L.-K.)

**Keywords:** CRISPR/Cas9, base editors, prime editors, human induced pluripotent stem cells, mRNA

## Abstract

In contrast to CRISPR/Cas9 nucleases, CRISPR base editors (BE) and prime editors (PE) enable predefined nucleotide exchanges in genomic sequences without generating DNA double strand breaks. Here, we employed BE and PE mRNAs in conjunction with chemically synthesized sgRNAs and pegRNAs for efficient editing of human induced pluripotent stem cells (iPSC). Whereas we were unable to correct a disease-causing mutation in patient derived iPSCs using a CRISPR/Cas9 nuclease approach, we corrected the mutation back to wild type with high efficiency utilizing an adenine BE. We also used adenine and cytosine BEs to introduce nine different cancer associated *TP53* mutations into human iPSCs with up to 90% efficiency, generating a panel of cell lines to investigate the biology of these mutations in an isogenic background. Finally, we pioneered the use of prime editing in human iPSCs, opening this important cell type for the precise modification of nucleotides not addressable by BEs and to multiple nucleotide exchanges. These approaches eliminate the necessity of deriving disease specific iPSCs from human donors and allows the comparison of different disease-causing mutations in isogenic genetic backgrounds.

## 1. Introduction

Since the discovery of human induced pluripotent stem cells (hiPSC) that can be generated by reprogramming somatic cells, hiPSCs-based disease modeling has been established as a versatile tool [1]. hiPSCs can be propagated indefinitely in vitro under growth conditions that maintain pluripotency and can also be differentiated into many different somatic cell types under appropriate culture conditions [2,3,4,5]. Typically, patient-specific hiPSC lines are generated through reprogramming of somatic cells isolated from a donor. These cells can then be applied for studying molecular and cellular disease mechanisms, as well as for drug screening [6,7,8] and for the development of possible cell therapies [9,10]. A major challenge of using patient-derived hiPSCs for modeling human diseases is their genetic diversity and differences in epigenetic memory [11,12]. This makes it difficult to compare data obtained from mutant iPSCs generated from a patient with other hiPSC lines generated from healthy individuals. Because each human has a unique genetic fingerprint, questions may arise as to whether observed phenotypic differences to the disease mutation are due to inherent differences in the genetic background of the cells. Elimination of genetic background effects would therefore be beneficial for loss-of-function and gain-of-function studies or drug screening. 

Thus far, attempts have focused on overcoming genetic background effects by introducing additional copies of the healthy gene into the genome of the diseased hiPSCs [10,13,14] or vice versa [15]. However, these methods are frequently ineffective because of the low transduction/transfection efficiency of iPSCs and non-physiological expression of added transgenes. Furthermore, co-expression of the wild-type and mutant genes in the same cell can lead to adverse effects. Finally, transgene expression from vectors can be unstable in iPSCs, leading to a decline of expression of the transgene over time [10,13,14]. Hence, transgene expression is not optimal to model or correct disease-causing mutations in human iPSCs. 

Recently developed genome editing tools are overcoming these limitations and can be used to correct the disease mutation or to introduce mutations of interest into wild-type (WT) cells. Especially the CRISPR/Cas9 system, a prokaryotic adaptive immune system repurposed for genome editing, has been widely adopted because of its unmatched simplicity and flexibility. Cas9 endonuclease in combination with a single guide RNA (sgRNA) can induce DNA double strand breaks (DSBs) at a specified genomic locus. The DSB can consecutively be repaired either by homology directed repair (HDR) or by non-homologous end joining (NHEJ) [16,17,18,19]. Consequently, disease-causing mutations have been corrected in human iPSCs when templates carrying the wild-type sequence were co-delivered together with the nuclease [20,21]. However, efficiency of HDR, which is dependent on an exogenous DNA template, is generally low [22]. In contrast, NHEJ is typically more efficient, but associated with random nucleotide insertions or deletions (indels), which are undesirable in the case of predefined desired nucleotide alterations. 

As an alternative to the conventional Cas9 system, so-called “base editors” (BEs) have recently been developed. These enzymes are able to edit specific bases in the genome without the induction of DSBs. By using a nickase version of Cas9 (nCas9) or dead Cas9 (dCas9) fused to a deaminase enzyme, these tools can exchange certain nucleotides in the genome, including the genome of iPSCs [23,24]. nCas9 fused to an APOBEC1 deaminase enzyme and to Uracil Glycosylase inhibitor (UGI) can effectively convert cytosine (C) into thymine (T) [25]. A similar system is the Target-AID, where nCas9 is fused to PmCDA1, a cytidine deaminase from sea lamprey [26]. To allow conversions of guanine (G) to adenine (A), Gaudelli et al. recently evolved an RNA adenine deaminase to accept DNA as a substrate [27]. Similar to Cas9 nuclease, BEs are guided to their target by sgRNAs and are able to introduce specific point mutations without generating DSB. Therefore, BEs avoid undesired DNA alterations, such as stochastic indels, large deletion, or genomic rearrangements [27]. However, BEs should also be used with caution since off-target effects on DNA and RNA have recently been described [28,29,30,31]. To minimize potential adverse effects, it has been suggested that the exposure time to the BEs should be reduced in the cells [32].

Base editing is currently restricted to changing single bases and only certain nucleotide changes can be achieved. The recently developed prime editors (PEs) offer more flexibility and can be used to convert any nucleotide or modify several bases at the same time [33]. However, this technology has thus far not been tested in human iPSCs. 

To investigate the correction of disease-causing mutations and the generation of different point mutations in an isogenic genetic background, we explored different forms of CRISPR genome editing in human iPSCs by delivering base editing as well as prime editing machineries as RNAs. This transient, DNA-free approach reduces off-target editing and the risk of spurious DNA integration.

## 2. Materials and Methods 

### 2.1. Lentiviral Vector Production

HEK 293T were maintained in DMEM medium supplemented with 10% (*v*/*v*) heat-inactivated fetal bovine serum (FBS), 2 mM glutamine, 100 U/mL penicillin, and 100 mg/mL streptomycin, at 37 °C with 5% CO_2_ in a humidified incubator. Lentiviral supernatants were produced by polyethylenimine (PEI) based transient co-transfection of HEK293T cells. Briefly, SEW vector (SFFVp-GFP), lentiviral gag/pol helper plasmids (psPAX2, Addgene#12260) and envelope plasmid VSV-G (pMD2.G, Addgene #12259) were transfected at a molar ratio of 3:1:1 by standard PEI transfection. Forty-eight hours post transfection, viral supernatants were harvested, sterile filtered (0.45-mmpore-size PVDF- membrane filter; Millipore, Burlington, MA, USA), and stored at −80 °C. SEW lentiviral titers were determined in serial dilutions of viral supernatant by transduction HEK293T cells followed by flow cytometry 4–5 days post-transduction.

### 2.2. In Vitro mRNA Transcription 

In vitro transcribed (IVT) mRNA was prepared from a PCR amplicon carrying the gene of interest, which were generated with the primer in Appendix A and Herculase II Fusion DNA Polymerase (Agilent, Santa Clara, CA, USA). IVT mRNAs were generated according to the manufacturer’s guidelines using the HiScribe T7 ARCA mRNA Kit (NEB, Ipswich, MA, USA) followed by purification using Monarch RNA Cleanup Kit (NEB, Ipswich, MA, USA). 

### 2.3. HEK293T Cells Transfection with mRNA

Cells were plated in 24-well format one day before transfection to reach 80% confluency at the time of transfection. 

For base editing, 1 μL Lipofectamine MessengerMAX (Invitrogen, Carlsbad, CA, USA) was diluted into 25 μL of Opti-MEM™ I Reduced Serum Medium (Thermo Fisher Scientific, Waltham, MA, USA), vortexed briefly, and incubated for 10 min at RT. Meanwhile, 1 pmol of BE mRNA was added to 25 μL of Opti-MEM, followed by addition of 5 pmol sgRNA. MessengerMAX solution was then mixed with mRNA/sgRNA sample and incubated for 5 min prior to addition to the cells.

For prime editing, 1 μL of Lipofectamine MessengerMAX was added to 25 μL of Opti-MEM medium, vortexed briefly, and incubated for 10 min at RT. To a separate test tube, 1 pmol of PE mRNA was added to 25 μL of Opti-MEM medium, followed by addition of 5 pmol pegRNA and 1.25 pmol sgRNA. The diluted transfection reagent was transferred to the tube containing PE/gRNAs complexes, followed by incubation at room temperature for 5 min and then added to the cells.

In each case, the entire solution was added to the cells in a 24-well plate and mixed by gently swirling the plate. Cells were analyzed 7–10 days post-transfection.

### 2.4. Flow Cytometry and Cell Sorting 

For flow cytometry, cells were washed and resuspended in PBS. Data acquisition was performed with a BD LSRFortessa flow cytometer (BD Biosciences, Franklin Lakes, NJ, USA) of the Flow Cytometry Core Facility, a core facility of CMCB at TU Dresden. Data were analyzed with BD FACSDiva software (BD Biosciences, Franklin Lakes, NJ, USA) or flowjo software (FlowJo LLC, Ashland, OR, USA). Cell sorting was performed in a BD FACSAria III flow cytometer (BD Biosciences, Franklin Lakes, NJ, USA) of the Flow Cytometry Core Facility, a core facility of CMCB at TU Dresden.

### 2.5. Reprogramming, hiPSC Culture and Characterization

AAVS1-Puro-CAG-eGFP targeted hiPSC (hPSCreg, BIHi001-A-2, Berlin, Germany) were obtained from the Core Facility Stem Cells at the Berlin Institute of Health (BIH) and cultured in Essential 8 Flex medium (Thermo Fisher Scientific, A2858501, Waltham, MA, USA) on truncated Vitronectin VTN-N (Thermo Fisher Scientific, A31804, Waltham, MA, USA), passaged using ReLeSR (StemCell Technologies, 05873, Vancouver, BC, Canada) as non-enzymatic clump passaging agent, and cryopreserved in mFreSR media (StemCell Technologies, 05855, Vancouver, BC, Canada).

Reprogramming, culturing, and characterization of CRTD2, CRTD3, and hSAMHD1-R290H+Q548X hiPSCs were performed at the Stem Cell Engineering Facility of the Center for Molecular and Cellular Bioengineering (CMCB) at TU Dresden using the CytoTune-iPS 2.0 Sendai Reprogramming Kit (Thermo Fisher Scientific, A16517, Waltham, MA, USA) according to the manufacturer’s guidelines for transduction. CRTD2 and CRTD3 hiPSCs were generated from MACS-sorted CD34+ cells from peripheral blood of a consented healthy donor (ethics vote at TU Dresden: EK 363112012). hSAMHD1-R290H+Q548X hiPSC were generated from patient peripheral blood mononoclear cells (PBMCs) (EK169052010 und EK386102017). Following transduction, the cells were cultured with ReproTeSR medium (StemCell Technologies, 05920, Vancouver, BC, Canada) on hES-qualified Matrigel (Corning, 354277, Corning, NY, USA) under standard conditions (5% CO_2_, 37 °C) for 18–21 days. Once hiPSC colonies were sufficiently grown, the clonal colonies were mechanically isolated and expanded in mTeSR1 medium (StemCell Technologies, 85850, Vancouver, BC, Canada) using ReLeSR and cryopreserved in mFreSR.

Pluripotency of the hiPSCs was confirmed using flow cytometry with Alexa Fluor 488-coupled anti-Oct3/4 (560253), PE-coupled anti-Sox2 (560291), V450-coupled SSEA-4 (561156), and Alexa Flour 647-coupled anti Tra-1-60 (all from BD Pharmingen, 560122) according to the manufacturer’s protocol at BD™ LSR II (BD Biosciences, Franklin Lakes, NJ, USA) from the Flow Cytometry Core Facility at CMCB at TU Dresden. 

The three-germ layer differentiation capacity of hiPSC was tested by embryoid body differentiation. Subconfluent hiPSC cultures were incubated in Collagenase Type IV (StemCell Technologies, 07909, Vancouver, BC, Canada), scraped from the plate producing big fragments and incubated for 8 days on low attachment plates followed by 8 days on gelatine-coated dishes in Advanced DMEM (Thermo Fisher Scientific, 12491023, Waltham, MA, USA) with 10% FBS (Thermo Fisher Scientific, 10270106, Waltham, MA, USA) that was supplemented for the first day of differentiation with 10 µM Y-27632 rho kinase inhibitor (StemCell Technologies, 72308, Vancouver, BC, Canada). Differentiated cells were fixed with 4% paraformaldehyde for 10 min, permeabilized for 15 min with 0.5% Triton X-100, and blocked for 30 min with 3% BSA/PBS at RT. For ectoderm, cells were stained with Anti-Tubulin, beta III TUBB3 (Sigma-Aldrich, MAB1637, St. Louis, MO, USA) and for mesoderm with Mouse anti-Actin (smooth muscle) SMA (Life Technologies, 180106, Carlsbad, CA, USA) primary antibody followed by Alexa Fluor 488 goat anti-mouse IgG (Thermo Fisher Scientific, A11001, Waltham, MA, USA) secondary antibody. Directed endoderm differentiation was performed with STEMdiff Trilineage Differentiation Kit (StemCell Technologies, 05230, Vancouver, BC, Canada) for 4 days according to manufacturer’s instructions, following fixation, permeabilization, blocking, and staining with MS mAb to SOX17 (3B10) primary antibody (Abcam, ab84990, Cambridge, UK) and Alexa Fluor 488 goat anti-mouse IgG. Nuclei were counterstained with NucBlue™ Fixed Cell ReadyProbes™ Reagent (Thermo Fisher Scientific, R37606, Waltham, MA, USA).

To test for chromosomal alterations, subconfluent, exponentially growing cultures from each hiPSC line and subclone were blocked for 4 h with 100 ng/mL Colchemid (Thermo Fisher Scientific, 15212012, Waltham, MA, USA). Single cells were harvested with TrypLE Express (Thermo Fisher Scientific, 12604013, Waltham, MA, USA), enlarged with 75 mM hypotonic KaryoMAX™ Potassium Chloride Solution (Thermo Fisher Scientific, 10575090, Waltham, MA, USA) for 20 min at 37 °C, fixed, and washed 3 times with 3:1 methanol:glacial acetic acid (Carl Roth, CP43.3, Karlsruhe, Germany and Merck, 1000631011, Darmstadt, Germany). Chromosome spreads were Giemsa stained and 20 metaphases of each sample were analyzed by G-banding [34] at the Institute of Human Genetics, University Clinics Jena, Germany.

### 2.6. Nucleofection

For generating single cells for nucleofection, hiPSCs were incubated 2 h with 10 µM Y-27632 rho kinase inhibitor, washed with DPBS (Corning, 21-031-CVR, Corning, NY, USA), dissociated with TrypLE Express and spun down (400 g, 5 min). Cell pellets of 4 × 10^5^ or 8 × 10^4^ cells were resuspended with 100 or 20 µL supplemented nucleofector solution from P3 Primary Cell 4D-Nucleofector^®^ X Kit L or S (Lonza, V4XP-3024 and V4XP-3032, Basel, Switzerland) with indicated amounts of mRNA, sgRNAs (Synthego or in vitro-transcribed with EnGen sgRNA Synthesis Kit, S. pyogenes, NEB, E3322S), single-stranded DNA oligos (Sigma-Aldrich, St. Louis, MO, USA), or Cas9-NLS protein (NEB, M0646M, Ipswich, MA, USA) and nucleofected with 4D-Nucleofector Core and X Unit (Lonza, Basel, Switzerland), program CB-150.

For eGFP mRNA transfections into CRTD2 and CRTD3, hiPSCs, 4 × 10^5^ cells in 100 µL were nucleofected with 3 or 7.5 pmol eGFP mRNA and cultured for 1 day in mTeSR1 with 10 µM Y-27632 rho kinase inhibitor before analysis at BD™ LSR II (BD Biosciences, Franklin Lakes, NJ, USA) of the Flow Cytometry Core Facility of CMCB at TU Dresden.

For ABE transfections into AAVS1-Puro-CAG-eGFP hiPSCs, 8 × 10^4^ cells in 20 µL were nucleofected with 0.6 pmol ABE mRNA and 100 pmol sgRNAs (Synthego). For PE transfections into AAVS1-Puro-CAG-eGFP hiPSCs, 8 × 10^4^ cells in 20 µL were nucleofected with 0.6 pmol PE mRNA, 75 pmol pegRNAs (Synthego) and 25 pmol sgRNA (Synthego). After transfection cells were replated in mTeSR1 with 10 µM Y-27632 rho kinase inhibitor for 1 day, before the medium was switched back to Essential 8 Flex for another 8-10 days until analysis at BD™ LSR II (BD Biosciences) of the Flow Cytometry Core Facility of CMCB at TU Dresden.

For HDR-based correction of hSAMHD1-R290H+Q548X hiPSCs, 30 pmol Cas9-NLS protein were pre-incubated for 20 min at RT with 31 pmol hSAMHD1-Q548H-sgRNA (EnGen sgRNA Synthesis Kit, NEB, Ipswich, MA, USA) and then mixed with 135 pmol 120 nt ss DNA oligo as HDR template and 2 × 10^5^ cells in 20 µL for nucleofection. 

For ABE-based correction of hSAMHD1-R290H+Q548X, 8 × 10^5^ hiPSCs in 100 µL were nucleofected with either 1 µg pCMV-ABE7.10 plasmid (Addgene, 102919, Watertown, MA, USA) and 2 µg of a pBR332-U6 sgRNA expression plasmid or 3 pmol ABE-puro mRNA and 93 pmol hSAMHD1-Q548H-sgRNA (EnGen sgRNA Synthesis Kit, NEB, Ipswich, MA, USA). After nucleofection, cells were seeded at limited dilution in mTeSR1 medium supplemented with CloneR (StemCell Technologies, 05888, Vancouver, BC, Canada) for 3 days. The cells were transiently selected using 0.5 µg/mL Puromycin (Thermo Fisher Scientific, A1113803, Waltham, MA, USA) for 24 h starting 24 h after nucleofection. Individual subclones were manually picked at Day 11 and expanded. One half of each picked colony was lysed for PCR amplification (Phusion high-fidelity PCR Master Mix with GC buffer, Thermo Fisher Scientific, F-532L) of hSAMHD1 exon 15 and Sanger Sequencing (Microsynth, Balgach, Switzerland). 

### 2.7. Transfection

For TransIT-LT1 transfection, the medium of AAVS1-Puro-CAG-eGFP hiPSC, usually cultured in Essential 8 Flex, was changed to mTeSR1 24 h before transfection. Per sample, 400 µL pre-warmed Opti-MEM™ I Reduced Serum Medium (Thermo Fisher Scientific, 31985062, Waltham, MA, USA) were mixed with 10 pmol GFP or mCherry mRNA and 10 µL TransIT^®^-LT1 (VWR, 731-0027) unless indicated otherwise, vortexed briefly, and incubated for 20 min at RT. The transfection mix was added to 6 wells pre-coated with Matrigel or VTN-N and incubated for 5 min at RT before addition of the cell suspension. CRTD2, CRTD3, or AAVS1-Puro-CAG-eGFP hiPSC were dissociated into a single cell solution using ReLeSR and repeated pipetting, resuspended in mTeSR1 with 10 µM Y-27632 rho kinase inhibitor and 2 × 10^6^ cells per well were seed on top of the transfection mix. Cells were cultured for 1 day before analysis at BD™ LSR II (BD Biosciences, Franklin Lakes, NJ, USA) from the Flow Cytometry Core Facility of CMCB at TU Dresden.

For MessengerMax transfection the hiPSCs were cultured in Cellartis DEF-CS 500 culturing system (Takara Bio, Shiga, Japan) according to the manufacturer’s recommendations. Typically, 1 million cells were seeded and transfected 16 h post transfection using 3.75 µL RNA MessengerMax reagent per well of a 6-well plate. We used 3.7 µg CBE-GFP or 5 µg ABE mRNA and always kept molar ratio of mRNA:gRNA at 1:5. The transfection media was removed 4 h post transfection. Ninety-six hours post transfection, cells were collected and genomic DNA was isolated. We amplified different *TP53* loci. These PCRs were then deep sequenced to reveal the editing efficiency. 

### 2.8. EdU Staining of hiPSCs under Nutlin Treatment

hiPSCs were treated with 5 µM Nutlin-3 or DMSO control for 48 h, and then labeled with 10 µM EdU for 2 h, using Click-IT EdU Flow Cytometry Assay Kit (Invitrogen, Carlsbad, CA, USA). After fixation and permeabilization, Click-iT reaction was performed according to the manufacturer’s recommendations. Finally, cells were stained with FxCycle Violet dye (Invitrogen, Carlsbad, CA, USA) for DNA content. Cell cycle analysis was performed by FACS (MACSQuant VYB, Miltenyi Biotec, Bergisch-Gladbach, Germany), measuring EdU-Alexa Fluor 488 for nucleotide incorporation and FxViolet for total DNA content. 

### 2.9. Statistical Analysis 

For statistical comparisons between groups, one-way ANOVA and Tukey’s multiple comparisons test were used as appropriate in conjunction with GraphPad Prism 10 software.

## 3. Results

### 3.1. Attempt for Correction of a Disease-Causing Mutation by HDR

Aicardi–Goutières syndrome (AGS) is a hereditary rare neuro-inflammatory disorder. Mutations in several genes can cause Aicardi–Goutières syndrome, including alterations in the *SAMHD1* gene [35,36]. To study the role of disease-causing mutations in *SAMHD1* in vitro, we generated hiPSCs from a patient with Aicardi–Goutières syndrome (AGS) (Appendix A). The cells are compound heterozygous and carry a missense mutation (c.869G > A; p.R290H) on one allele, whereas the other allele harbors a nonsense mutation (c.1642C > T; p.Q548*) (Appendix A). To create isogenic control lines, we attempted correcting the nonsense mutation. For this purpose, we designed an sgRNA and a corresponding repair template (Appendix A). The patient iPSCs were nucleofected with Cas9-NLS protein, sgRNA, and a 120-nt-long ssDNA oligonucleotide with 59-nucleotide homology upstream, and 60-nucleotide homology downstream of the mutation, respectively. Seventy-two single clones were picked and analyzed for the correction by PCR and sequencing. Unfortunately, we were not able to find any clone with the desired correction (Appendix A). A possible explanation for the failure is that the only conceivable sgRNA cuts at a distance of 12 base pairs away from the actual mutation. This distance might be suboptimal to promote HDR with the employed repair oligo. Therefore, we were unable to correct the p.Q548* mutation in the *SAMHD1* gene in patient iPSCs utilizing CRISPR/Cas9 nuclease in combination with a HDR repair template.

### 3.2. mRNA-Mediated Base Editing in HEK293T Cells

To investigate, whether the mutations in the *SAMHD1* gene could potentially be corrected by means of base editing, we first tested base editing in HEK293T cells. Recently, extensive off-target editing has been described for BEs [28,31]. These off-target events are particularly relevant for iPSCs. To minimize these effects, short time expression of the BE proteins has been recommended [32], which is difficult to achieve with transfection of plasmid DNA into cells. However, mRNA delivery has been shown to limit the expression of other genome editing enzymes, including Cas9 nuclease [37,38]. To test the feasibility and efficiency of base editing via mRNA transfection, we utilized a HEK293T reporter cell line with single copy integration of the green fluorescent protein (GFP) gene. We transfected these reporter cells with adenine base editor (ABE) or cytosine base editor (CBE) mRNA (Figure 1A) together with four different sgRNAs targeting the GFP gene (Figure 1B). Two of these sgRNAs target the start codon of the GFP gene with the aim to inactivate GFP expression. The third sgRNA targets the amino acid tyrosine at position 66 and changes it into a histidine after successful editing, leading to conversion of GFP into blue fluorescent protein (BFP). The fourth sgRNA targets glutamine 158, changing it into a stop codon. With all four sgRNAs and both base editors, we were able to induce the intended editing events, with efficiencies ranging from 13% to 47% (Figure 1C,D). To confirm the presence of the desired modifications, we sorted the edited cells and analyzed their GFP gene by sanger sequencing (Figure 1C). The sequencing data revealed that all expected nucleotides were changed in the anticipated manner, demonstrating that the short-term expression after mRNA transfection together with the sgRNAs is sufficient and effective to change specific nucleotides in mammalian genomes. However, for the ABE-Y66H as well as for the CBE-M1* editing, additional bystander nucleotides in the target site were also altered, although with low frequencies (Figure 1C). Even though the bystander editing frequency for ABE-Y66H was lower than the efficiency of the desired edit (43%), the change led to an additional amino acid p.L64P substitution, which inactivated GFP expression in ~3.0% of cells.

### 3.3. mRNA Delivery into hiPSCs

To optimize transfection of mRNA into hiPSC, we transfected different hiPS cell lines with EGFP (Figure 2A), mCherry or TagBFP mRNA with different transfection reagents as well as different cultivation conditions. We also tested forward as well as reverse transfection protocols (Appendix A). Interestingly, we observed a profound difference of transfection efficiencies using DEF-CS versus mTeSR1 and Essential 8 Flex culture conditions. Apparently, cultivation in mTeSR1 and Essential 8 Flex interfered with lipofection of the mRNAs into the cells. In contrast, very high transfection efficiencies were achieved in the DEF-CS culture system and forward transfection with Lipofectamine MessengerMAX, in which we recorded >95% transfection rates of the cells (Figure 2B,C and Appendix A). To investigate whether mRNAs could be delivered into iPSCs cultured in mTeSR1 and Essential 8 Flex by other means, we turned to nucleofection. Indeed, nucleofection of mRNAs under optimal conditions reached up to 90% (Figure 2C), indicating that this approach is an alternative when hiPSCs are cultured in mTeSR1 and Essential 8 Flex. However, nucleofection was associated with higher toxicity and cell death when compared to transfections with Lipofectamine MessangerMAX (data not shown). 

### 3.4. BE-Mediated Conversion of GFP into BFP in hiPS Cells

To test whether BE-mRNA co-delivery with sgRNAs could also produce efficient changes in the genome of hiPSC, we nucleofected the ABE-mRNA and the GFP-Y66H sgRNA into hiPSCs carrying two copies of the GFP gene in the AAVS1 locus (AAVS1-eGFP). From 8 to 10 days after transfection, the conversion of GFP-to-BFP was analyzed by flow cytometry (Figure 3A). Remarkably, the fraction of edited cells that had turned GFP into BFP was up to 80% (Figure 3B,C), demonstrating efficient base editing with mRNA in hiPSCs. The two copies of the GFP gene allowed us to record the conversion efficiency mimicking the biallelic state of most genes in the genome. Around 40% of the cells showed expression of BFP only, indicating that both copies of the GFP gene had been converted. In contrast, around 40% of the cells expressed both GFP and BFP (Figure 3B,C), consistent with successful editing of only one allele. Similar to experiments performed in HEK293T cells, bystander editing with the GFP-Y66H sgRNA was also observed, albeit with lower frequencies of ~0.5%. 

### 3.5. Repair of a Disease-Causing Mutation in Patient-Derived hiPSCs

Based on the promising results with mRNA-mediated base editing of GFP in HEK293T and hiPSCs, we went back to the patient-derived *SAMHD1* mutant hiPSCs in which we had failed to correct the mutation with Cas9 nuclease in combination with HDR-mediated oligo repair (Appendix A). While for the p.R290H mutation no sgRNA with the mutation in the editing window could be identified, the sole sgRNA spanning the p.Q548* mutation that was used in the unsuccessful HDR attempt seemed to be optimally oriented for base editing (Figure 4A). We nucleofected these hiPSCs with ABE mRNA and an sgRNA or with ABE- and sgRNA-plasmid to correct the mutation and plated the cells so that individual clones could grow in mTeSR1 medium. From the plasmids experiment, we did not obtain any homozygote repaired clones. Out of 48 clones, 10 clones died, 21 clones were unedited, and 17 clones were mixed (Appendix A).

From the mRNA and sgRNA experiment, of 21 resulting subclones, five (24%) died during expansion and six (29%) were unmodified as they still harbored the original heterozygous mutation. Four clones (19%) were fully corrected as intended and the residual six clones (29%) showed inconclusive results representing a mixture of unmodified (heterozygous mutated) and corrected alleles (Figure 4B). Importantly, none of the analyzed ABE-transfected subclones showed any undesired bystander modifications or indels that were observed at a frequency of approximately 3% with the unsuccessful HDR approach (Appendix A) using the same sgRNA. These data demonstrate that the ABE mRNA approach not only corrected the mutation more efficiently than HDR, but also that it worked with high accuracy. 

Three of the successfully corrected clones and the parental patient line were extensively characterized and normal expression of pluripotency markers was confirmed by flow cytometry (Appendix A). A three-germlayer differentiation potential (Appendix A) and an intact karyotype (Appendix A) showed that ABE editing did not impair pluripotency or genomic stability of the hiPSCs. Short tandem repeat (STR) analysis confirmed the genetic identity of the parental line and the subclones (data not shown). Furthermore, we characterized the three corrected clones by mRNA-Seq to detect potential off-targets. No significant differences could be detected between the parental line and the subclones (data not shown).

### 3.6. Generation of Isogenic hiPSCs with Different TP53 Mutations

Encouraged by the successful *SAMHD1* p.Q548* correction in the patient-derived hiPSCs, we wanted to investigate whether the introduction of disease-causing mutations into WT hiPSCs would also be feasible using BE mRNA transfections. A positive outcome of these experiments might remove the necessity to generate disease specific hiPSCs from human donors and would allow the generation of isogenic panels of diverse disease-causing mutations in the same healthy control line to prevent false positive or negative results caused by different genetic backgrounds. 

Large-scale DNA sequencing efforts have revealed the mutational landscape of cancer mutations and more than 300 cancer driver genes have been identified thus far [39]. However, the functional role of the individual mutations in these cancer driver genes is currently not well understood. In the most frequently mutated cancer gene *TP53* alone, more than 5000 recurrent mutations have been described (https://cancer.sanger.ac.uk/cosmic/gene/analysis?ln=TP53). To decipher the biology of these different mutations, it would be advantageous to compare them in an isogenic background. Approximately 75% of the p53 mutations in cancer are missense or nonsense mutations, which makes the *TP53* gene ideal to target with base editor systems [40,41]. To test feasibility, we designed nine sgRNAs for ABE or CBE to generate point mutations in the TP53 gene within hiPSCs (Figure 5A,B). The target mutations were chosen from the COSMIC database (https://cancer.sanger.ac.uk/cosmic/gene/analysis?ln=TP53) and represent recurrent mutations in cancer. Seven sgRNAs were designed to introduce missense mutations, while two sgRNAs were intended to introduce nonsense mutations into the *TP53* gene. We transfected the sgRNAs separately with ABE or CBE mRNA into WT hiPSCs and four days after transfection, isolated genomic DNA from the cells to monitor the genome editing events by deep sequencing. We observed different editing rates ranging from 1–90% depending on the sgRNA and BE system employed (Figure 5C). Surprisingly, all experiments with ABE showed higher editing rates than CBE experiments (Figure 5C). ABE-C141R and CEB-C141Y had the same sgRNA and edited the same amino acid; ABE changed the codon TGC > CGC (p.C141R), while CBE performed TGC > TAC (p.C141Y, editing in antisense strand). Although the sgRNAs were identical between the two systems, we measured six times higher editing rate with the ABE (52.4% vs. 8.8%, Figure 5C). Furthermore, the deep sequencing data also revealed that the cells transfected with the CBE mRNA had additional base changes (Appendix A), indicating that CBE editing is not as accurate as ABE editing. Since a higher cell toxicity was observed with CBE compared to ABE mRNA, we speculate that this might have contributed to lower observed editing rates as cells with high CBE mRNA content might have died shortly after transfection due to off-target effects. Hence, genome editing in hiPSCs was more efficient and more reliable with the ABE system. 

To functionally investigate different mutations in p53 within these hiPSCs, we treated the ABE edited cell populations with the *MDM2* inhibitor Nutlin-3, a drug that interrupts the interaction between p53 and the ubiquitin-ligase *MDM2* [42]. Nutlin-3 stabilizes p53, thus causing strong cell cycle arrest in WT, which is abrogated in mutant p53 cells [43]. To investigate whether Nutlin-3 treatment affects cells harboring the p53 mutations p.C141R, p.Y163H, and p.H193R, we incubated these cells and WT cells with 5 µM Nutlin-3 or DMSO as control. Forty-eight hours post treatment we performed cell cycle analyses, as measured by dual EdU-FxCycle Violet labeling. Consistent with a role of Nutlin-3 in stabilizing p53, we observed a marked increase in G2 arrested cells in the WT cells. In contrast, cells carrying the p53 mutations were more resistant to this treatment (Figure 5D), unmasking their oncogenic potential. Interestingly, we observed a significant difference in G2 arrested cells for the p.Y163H mutation in comparison to the cells carrying the p.C141R and p.H193R mutations, indicating that the p.Y163H mutation renders the cells more resistant to Nutlin-3 treatment than the other two mutations. 

### 3.7. Prime Editing in hiPS Cells

Because base editing is currently limited to A→G (ABE) or C→T (CBE) modifications and depends on an appropriate sgRNA for a successful genome alterations, we finally wanted to explore the recently developed prime editing approach [33] for genome editing in hiPSCs. To test if mRNA-mediated prime editing can also work efficiently in hiPS cells, we generated an mRNA from PE and tested three different pegRNAs (Figure 6A,B), which differed in nucleotide length of the primer binding site (PBS). Similar to the previous experiment where we converted GFP-to-BFP, here we designed pegRNAs to change two bases. This time, we converted the amino acid tyrosine at position 66 in the GFP gene into tryptophan (p.Y66W), which converts GFP into cyan fluorescent protein (CFP) by changing the codon TAC > TGG. We first tested prime editing in HEK293T-EGFP, which harbors a single copy of the GFP gene. Co-transfection of target cells with PE mRNA, sgRNA, and pegRNA resulted in a GFP-to-CFP conversion efficiency up to 6% (Appendix A). Interestingly, there was a difference between each pegRNA depending on the length of the PBS (Figure 6B). While the pegRNA11 with the shortest PBS (11 nt) showed the highest editing (6.5%), pegRNA15 with the longest PBS (15 nt) showed the lowest editing rate (0.3%) (Appendix A). 

Repeating the experimental setup in the AAVS1-eGFP hiPS cells, we were able to modify GFP-to-CFP in up to 7.5% of the cells (Figure 6C,D). Noteworthy, we were only able to quantify cells where both GFP alleles were edited, because the GFP fluorophore is also excited in the CFP channel. Therefore, it was not possible to differentiate heterozygously edited cells from unedited cells. Surprisingly, prime editing was in generally more efficient in hiPSC compared to HEK293T cells with the same pegRNAs. Even the less efficient pegRNA13 was 2.5 times more efficient in hiPSC than in HEK293T cells (4.4% vs. 1.7%) (Figure 6C and Appendix A). To ultimately confirm successful editing, we sorted CFP positive cells from AAVS1-eGFP hiPSCs and sequenced over the edited area. The sequencing results revealed the expected changes of the TAC codon to the desired TGG codon (Figure 6E).

## 4. Discussion and Conclusions

In the last few decades, hiPS cells have become an invaluable tool for human disease modeling. hiPS cells are particularly critical for studying rare monogenic diseases, because of limited access to affected tissues, such as the central nervous system (CNS) or muscle. The ability of iPSCs to differentiate into virtually any desired cell type makes iPSCs an ideal source for the generation of cells affected by these diseases. However, genetic background can impact the phenotypic outcome of experiments. Therefore, it is preferable to use cells with the same genetic and epigenetic background as a disease model and as a control. Methods involving CRISPR/Cas9 and HDR technology offer a plausible way for precise modifications of iPSCs, but the technology suffers from low efficiency and off-target effects. Therefore, identifying correctly modified iPSC requires extensive screening of clones [20,21,22]. 

BEs and PEs offer an alternative route to modify the genome of iPSCs. Our results indicate that the correction of disease-causing mutations can be more effective when BEs are employed compared to nuclease-based HDR. The high editing frequencies should reduce efforts to identify clones with the desired alteration. A problem emerging with BEs is DNA and/or RNA off target editing [28,29,30,31]. These side effects increase the risk of unwanted alteration and iPSCs might be particularly vulnerable to this, as expression changes due to cross editing of RNAs could lead to loss of pluripotency and/or differentiation potential. Thus far, plasmids have primarily been used to deliver the BEs into cells, which leads to long-term expression of the enzymes. Limiting the expression window of editing enzyme has been proposed to minimize off-target effects [32]. Our results indicate that delivery of BEs and PEs via mRNA transfection is an efficient way for short-term expression of the systems without compromising pluripotency of the cells.

While efficient and specific, certain off-target base editing is currently difficult to avoid. A fraction of ABE-Y66H and almost all CBE-M1* modified cells showed changes in two bystander nucleotides within the editing window in addition to the desired mutation. These results are in agreement with previous reports [25,44,45] confirming the activity of BEs across the whole editing window. Thus, considering the surrounding nucleotides of each locus is mandatory when planning and designing BE experiments.

Optimization of the transfection efficiencies using different hiPSC conditions revealed sharp differences, depending on the culture conditions employed. Although nucleofection has found widespread use as a method for CRISPR/Cas9 DNA delivery in hiPSCs, it is usually associated with high cell stress [46]. In our experiments, toxicity was also relatively high when mRNA was used during electroporation. Nevertheless, nucleofection was the most efficient mRNA transfection method tested in standard mTeSR1 and Matrigel culture conditions. Interestingly, the highest efficiency and lowest cell toxicity was achieved with the DEF-CS culture system and forward transfection with Lipofectamine MessengerMAX, indicating that this combination is optimal for efficient and non-toxic delivery of mRNA into hiPSCs. Using these conditions, we were able to introduce a mutation in up to 90% of the cells. Notably, the editing rates were higher in hiPSCs compared to HEK293T cells, while the bystander editing was about ten times lower. Altogether, lipofection using DEF-CS culture conditions appears to be the most recommendable way for mRNA-mediated base editing in hiPSCs.

We demonstrated the utility of base editing in hiPSCs by correcting a heterozygous mutation present in the *SAMHD1* gene in patient-derived hiPSCs and by generating nine isogenic cell lines with different *TP53* mutations. Observed modification efficiencies were high enough to enable rapid detection of clones with the desired mutations in almost all cases. The RNA-Seq results obtained indicate that off-target effects are rare when using ABE and that the hiPSCs remained pluripotent with an intact karyotype. At the same time, the use of isogenic iPSCs provided a strong basis to investigate the observed differences in Nutlin-3 resistance mediated by different *TP53* mutations. This approach should make it feasible to profile different mutations in disease-causing genes, which might ultimately help developing targeted treatment for specific mutations.

Although BEs are an effective tool for genome engineering, the scope of mutations they can introduce is limited. BEs can currently only convert C→T (CBE) or A→G (ABE) and are not able to introduce insertions or deletions. We therefore tested if the recently published PE system can be employed in hiPSCs utilizing PE mRNA. Indeed, we were able to convert GFP-to-CFP, which required changing two nucleotides. This conversion TAC→TGG (Y66W) cannot be achieved using current BEs. In total, we tested three pegRNAs, each having different editing efficiencies. The rules for designing the optimal length for pegRNAs are still emerging and the results presented here should be helpful to improve future design criteria. 

Taken together, our results support that BE as well as PE can be delivered as mRNA for efficient modification of hiPS cells. The biggest advantage of mRNA delivery is the possibility to express the editing enzymes transiently, thus minimizing the risk of off-target events and other unwanted effects that have been described for DNA delivery. Furthermore, because no DNA is delivered with this approach, the risk of unintended DNA integration (vector or template) in the genomic DNA is avoided [47]. While impressive for the engineering of hiPS cells as well as other cell lines, additional development is needed for therapeutic applications. In particular, the bystander editing (by BE) and indels formation (by CBE and PE) have to be solved before considering BEs and PEs for therapeutic applications. However, improvements of BEs have already been described [48,49] and it is foreseeable that improvements in the fidelity of PEs will follow soon.

## Figures and Tables

**Figure 1 genes-11-00511-f001:**
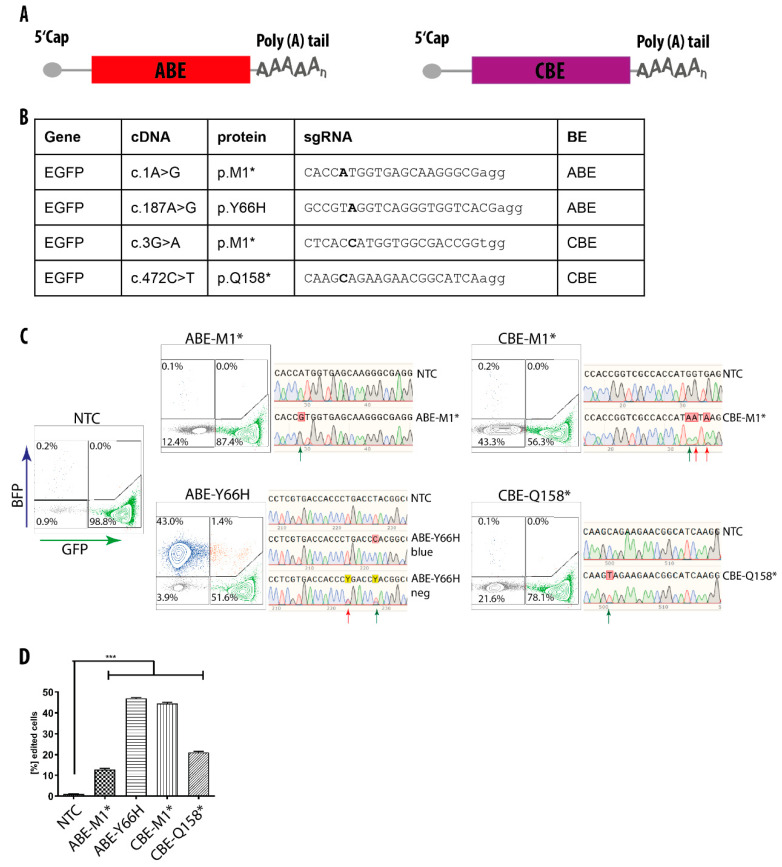
Base editing in HEK293T cells via mRNA transfection. (**A**) Schematic illustration of adenine and cytosine base editor (ABE and CBE) mRNAs. (**B**) GFP targeting sgRNAs. The introduced changes on DNA and protein levels, as well as the gRNA sequences and the employed BE are shown. (**C**) FACS profiles of HEK293T cells 10 days post co-transfection of indicated BE mRNAs and sgRNAs. The corresponding sequence reads recovered from sorted cells are shown to the right, and the green arrow indicates the expected change. Note the bystander mutations (indicated as red arrows) detected in the CBE-M1* fraction and in the ABE-Y66H GFP negative (neg) fraction. (**D**) Quantification of GFP edited cells after treatment. Results are represented as means ± SD of three independent experiments. NTC, non-targeting control. *** *p* < 0.001.

**Figure 2 genes-11-00511-f002:**
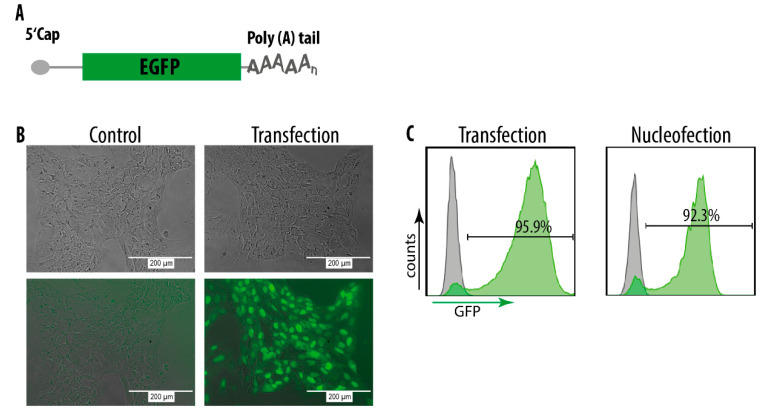
mRNA delivery into hiPS cells. (**A**) Schematic illustration of EGFP mRNA. (**B**) Fluorescent microscopy images of hiPS cells 24h post transfection with EGFP mRNA in the DEF-CS culture system with Lipofectamine MessengerMAX^TM^ transfection reagent. The upper panels show brightfield images and the lower panels show fluorescent recordings. (**C**) FACS profiles of hiPS cells 24h post transfection or nucleofection with EGFP mRNA. Grey = control cells (non-transfected).

**Figure 3 genes-11-00511-f003:**
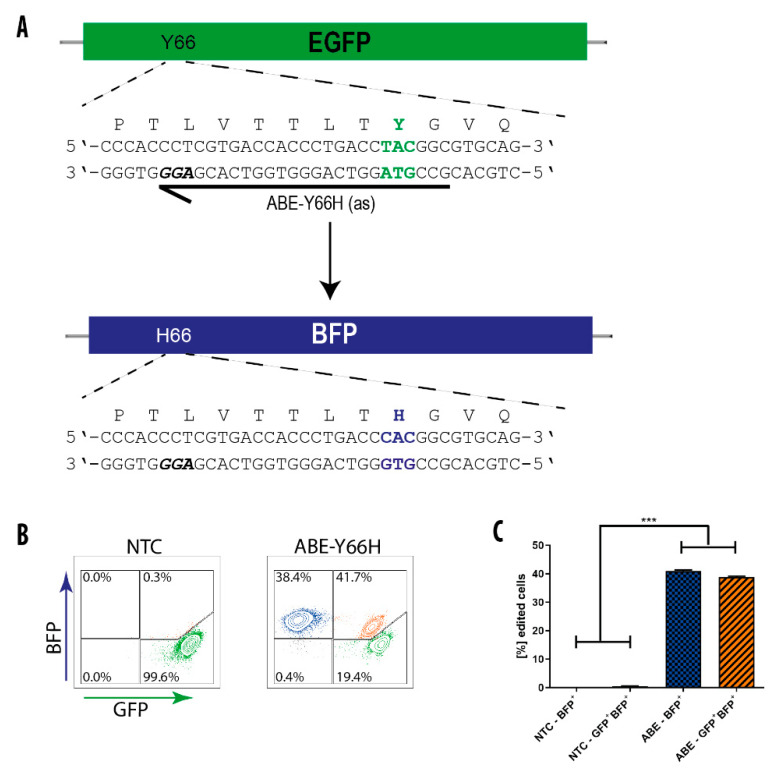
ABE conversion of GFP-to-BFP in hiPS cells. (**A**) Schematic illustration of the GFP-to-BFP conversion. The employed sgRNA is high-lighted by an arrow, with the protospacer-adjacent motif (PAM) sequence shown in bold italicized and the codon to be altered shown in bold green. The changed amino acids are depicted from green to blue, respectively. (**B**) FACS profiles of AAVS1-eGFP hiPS cells 10 days post co-transfection with ABE mRNA alone (NTC) or in combination with the GFP-sgRNA-Y66H. Note that AAVS1-eGFP hiPS cells harbor two copies of the GFP gene, explaining cells that are green and blue fluorescent (shown in orange). (**C**) Quantification of GFP edited cells after transfection. Results are represented as means ± SD of three independent experiments. *** *p* < 0.001.

**Figure 4 genes-11-00511-f004:**
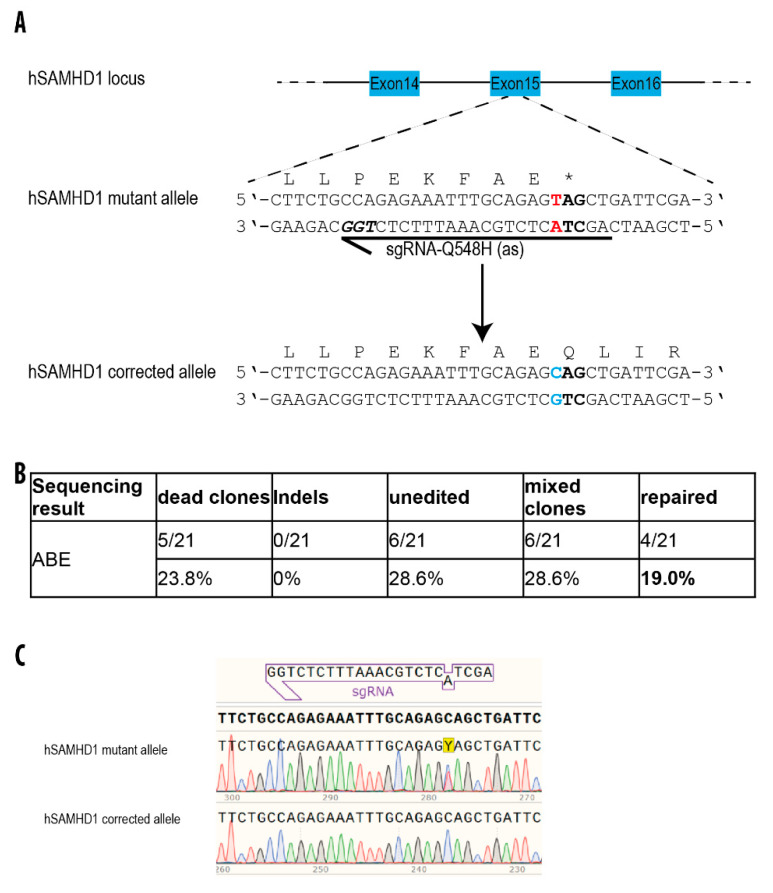
Repair of a *SAMHD1* mutation in patient-derived hiPS cells. (**A**) Schematic illustration of the relevant region of the *SAMHD1* locus. The sgRNA-targeting sequence is shown as an arrow, with the protospacer-adjacent motif (PAM) sequence indicated in bold italicized. The mutated codon is indicated in bold with the mutated nucleotide shown in red. (**B**) Editing rate of the *SAMHD1* gene utilizing the ABE system. Absolute numbers and percentages of investigated clones are provided. The percentage of correctly repaired clones is shown in bold. Mixed clones have still a residual pick of initial nucleotide in the Sanger sequencing. Almost 30% of the clones were unedited and no indels were detected. (**C**) Sanger sequencing data showing the heterozygous c.1642C > T (Q548X) mutation in the patient hiPSC line (top) and the ABE-corrected locus (bottom). The “Y” boxed in yellow highlights the nucleotide that was changed.

**Figure 5 genes-11-00511-f005:**
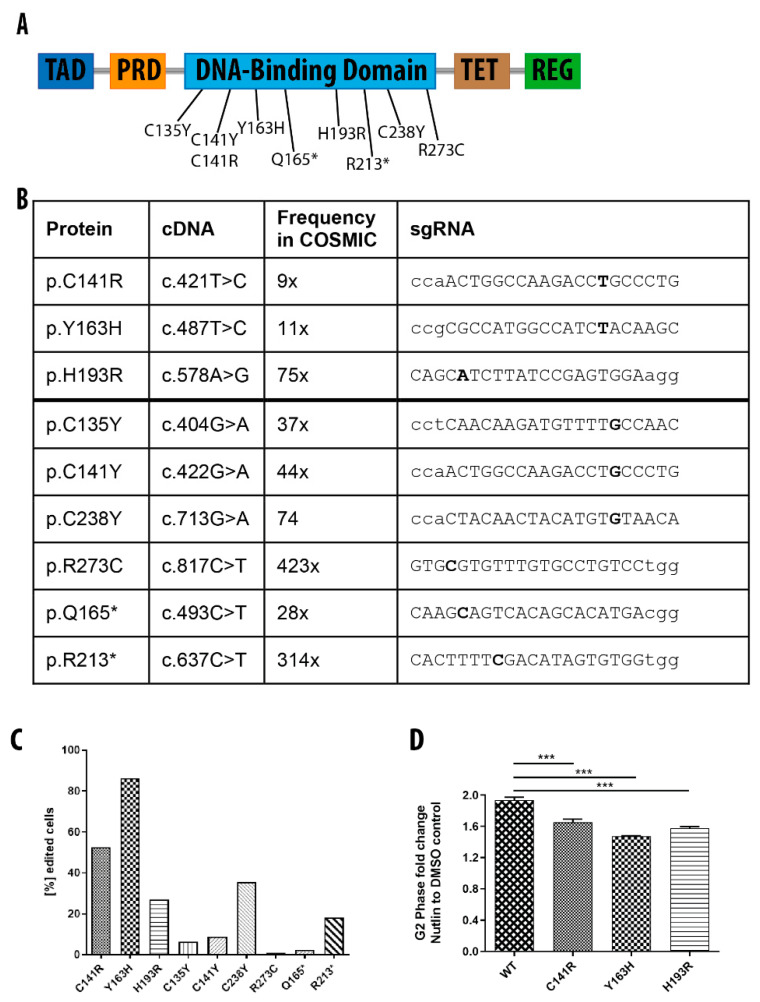
BE-mediated generation of isogenic hiPSCs carrying TP53 mutations. (**A**) Schematic illustration of the *TP53* gene with its addressed mutations labeled. TAD, transcriptional repression domain; PRD, proline- rich domain; TET, tetramerization domain; REG, basic C-terminal regulatory domain. (**B**) Important features of the selected *TP53* mutations. The introduced mutations, frequencies of the mutations in the cosmic database and the sequence of the employed sgRNAs are shown with nucleotides in bold highlighting the positions to be changed). (**C**) Quantification of editing efficiencies based on deep sequencing results of the *TP53* gene. (**D**) Quantification of cells in G2 after Nutlin-3 treatment in indicated cell lines. Results are represented as mean ± SEM of three independent experiments. *** *p* < 0.001.

**Figure 6 genes-11-00511-f006:**
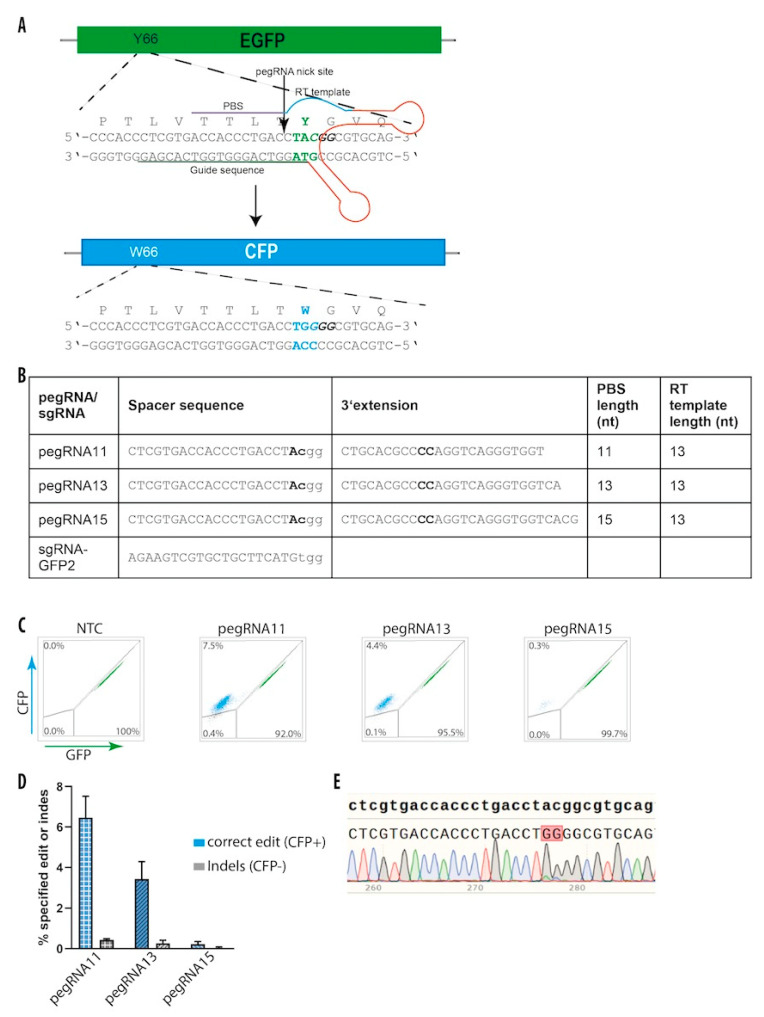
Conversion of GFP-to-CFP in hiPS cells with prime editing. (**A**) Schematic illustration of the GFP-to-CFP conversion. The employed pegRNA is high-lighted by a line with important features highlighted. PBS, primer binding site; RT, reverse transcriptase. The PAM sequence is shown in bold italicized. The position where Cas9 introduces a nick is indicated by an arrow. The changed codon and amino acids are depicted in green and blue, respectively. (**B**) Features of the employed pegRNAs and sgRNA for the GFP-to-CFP conversion. Nucleotides to be changed are depicted in bold. PBS, primer binding site; RT, reverse transcriptase. (**C**) FACS profiles of hiPS cells 14 days post transfection with indicated PEs. Percentages of the gated fractions are shown. (**D**) Quantification of editing efficiencies based on the FACS results. Results are represented as mean ± SEM of three independent experiments. (**E**) Sanger sequencing data from CFP positive sorted cells. Converted nucleotides are shown in a red box.

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
