# Peer review of "Efficient Generation and Correction of Mutations in Human iPS Cells Utilizing mRNAs of CRISPR Base Editors and Prime Editors"

_genes, 2020, doi:10.3390/genes11050511_

Round 1
Reviewer 1 Report
Several important have been developed since the discovery of the discovery of ground-breaking CRISPR/Cas system, and some of its most recent developments are the CRISPR-BE and CRISPR-PE methods enabling desired nucleotide exchanges in genomic sequences without generating double strand breaks. Another recent discovery that revolutionized biomedicine, the iPSCs, opens unprecedented ways to model human diseases in the dish, which strongly relies on generation of isogenic iPSC lines. The manuscript by Surun et al. “Efficient generation and correction of mutations in human iPSC cells utilizing mRNAs of CRISPR base editors and prime editors” represents a successful implementation of the CRISPR-BE and CRISPR-PE methods towards precise and efficient gene editing in human iPSCs. It gives an extremely useful practical guidance to researches dealing with disease modeling using human iPSCs. I think this is technically sound, well-performed, well-written and accomplished work that should be published in Genes without unnecessary delays.
Minor critics:
- «Interestingly, cells harboring the p.Y163H mutation showed hardly any increase in G2 arrested cells after the treatment, while for the cells carrying the p.C141R and p.H193R mutations, some cell cycle arrest in G2 was still measurable…» – the statement is exaggerated, considering that Nutlin/DMSO ratio p.Y163H is 1.4, whereas the ration for C141R, H193R is just slightly higher (1.5-1.6).
-Figure 6A: Looks like the pegRNA is not incorrectly depicted. It gives impression that the spacer-sequence anneals to the upper strand, which is not correct.
-Figure 6B and corresponding text: PBS length is not 11, 13, and 15 but 13, 15, and 17, respectively. The length is defined by the ribonucleotides complementary to the upper strand. Please double-check.
-Page 1: “stem cells (hiPSC) could be generated” change to “stem cells (hiPSC) that could be generated”
-Page 7: “Similar, to experiments performed in HEK293T cells” remove comma.
-Page 13: “Surprisingly, prime editing in hiPSC was in generally more efficient than in HEK293T”- remove in.
Author Response
Reviewer1:
Comments and Suggestions for Authors
Several important have been developed since the discovery of the discovery of ground-breaking CRISPR/Cas system, and some of its most recent developments are the CRISPR-BE and CRISPR-PE methods enabling desired nucleotide exchanges in genomic sequences without generating double strand breaks. Another recent discovery that revolutionized biomedicine, the iPSCs, opens unprecedented ways to model human diseases in the dish, which strongly relies on generation of isogenic iPSC lines. The manuscript by Surun et al. “Efficient generation and correction of mutations in human iPSC cells utilizing mRNAs of CRISPR base editors and prime editors” represents a successful implementation of the CRISPR-BE and CRISPR-PE methods towards precise and efficient gene editing in human iPSCs. It gives an extremely useful practical guidance to researches dealing with disease modeling using human iPSCs. I think this is technically sound, well-performed, well-written and accomplished work that should be published in Genes without unnecessary delays.
Reply: We thank reviewer #1 for the positive feedback and the constructive suggestions that we have addressed as outlined below:
Minor critics:
- «Interestingly, cells harboring the p.Y163H mutation showed hardly any increase in G2 arrested cells after the treatment, while for the cells carrying the p.C141R and p.H193R mutations, some cell cycle arrest in G2 was still measurable…» – the statement is exaggerated, considering that Nutlin/DMSO ratio p.Y163H is 1.4, whereas the ration for C141R, H193R is just slightly higher (1.5-1.6).
Reply (page 14, line 5): We have tuned down the statement. The sentence now reads: “Interestingly, we observed a significant difference in G2 arrested cells for the p.Y163H mutation in comparison to the cells carrying the p.C141R and p.H193R mutations, indicating that the p.Y163H mutation renders the cells more resistant to Nutlin-3 treatment than the other two mutations.”
-Figure 6A: Looks like the pegRNA is not incorrectly depicted. It gives impression that the spacer-sequence anneals to the upper strand, which is not correct.
Reply (page 16): We have improved the schematic presentation of Fig. 6A and now describe the different parts of the pegRNA in the illustration.
-Figure 6B and corresponding text: PBS length is not 11, 13, and 15 but 13, 15, and 17, respectively. The length is defined by the ribonucleotides complementary to the upper strand. Please double-check.
Reply (page 16): For the counting of the PBS length, we followed the specification proposed by Liu et al. (PMID: 31634902), which counts the bases upstream of the nick site. According to this specification, the PBS length is 11, 13 and 15.
-Page 1: “stem cells (hiPSC) could be generated” change to “stem cells (hiPSC) that could be generated”
Reply (page 1 line 1): We corrected this sentence, thank you.
-Page 7: “Similar, to experiments performed in HEK293T cells” remove comma.
Reply (Page 10, line 13): We corrected this sentence, thank you.
-Page 13: “Surprisingly, prime editing in hiPSC was in generally more efficient than in HEK293T”- remove in.
Reply (page 14, line 30): We corrected this sentence, thank you.
Reviewer 2 Report
Overview
Surun et al. had utilized the gene modification strategy using the base editors (BEs) or the prime editors (PEs) with the combination of mRNA transfection to generate precise point mutations in human iPSCs. Using BEs, the authors successfully showed the single base alternation in hiPSCs with high efficiency, which was verified with fluorescent expression level and the Sanger sequencing. Furthermore, human disease-causing mutation in isogenic hiPSCs was generated with BEs, suggesting the possible experimental scheme to investigate the disease without genetic background effects. Lastly, genetic modification approach with PEs was introduced for hiPSCs, which allows more flexible and simultaneous multiple base alternation. The paper aimed to explore the feasibility of using BEs or PEs in hiPSCs, and the authors illustrated that such genetic modification strategy works well in hiPSCs compared to the conventional Cas9 system. However, more thorough examination into the usage of BEs or PEs in hiPSCs would have made the paper more resourceful.
Specific comments
- In this paper, the conventional CRISPR/Cas9 system was shown to perform less efficient than newly introduced systems in hiPSCs, yet the experimental data for the previous CRISPR/Cas9 system was weakly demonstrated. For example, one of the major improvements is introduction of in vitro transcribed BE mRNA instead of BE-expressing plasmid or protein. At least one example of direct comparison between BE mRNA and BE plasmid in the same setting in terms of “off-target effect” need to be required. Also, the references and comparisons to the other papers would be very helpful.
- The comparison of transfection efficiency among the different types of media for hiPSCs transfection implied that the culture condition should be carefully considered. Since the paper intended to focus on the usage of new genetic alternation technologies in hiPSCs, the comparison should be made more precisely. Additionally, the insights into why different types of media resulted in varying transfection efficiency would benefit the readers.
- Statistical methods or p-values should be noted when statistical comparison was made (e.g. Figure 1D).
- Overall, the portion of fluorescently labeled cells was very high when using BEs for gene editing, indicating the highly efficient rate of transfection. In addition, the sequencing data also informed that number of edited cells is greater. In order to convey that the new technologies allow the greater transfection efficiency in hiPSCs, the quantitative results should be compared to the previous research, in which various types of gene editing technology were used for hiPSCs.
Author Response
Reviewer2:
Comments and Suggestions for Authors
Overview
Surun et al. had utilized the gene modification strategy using the base editors (BEs) or the prime editors (PEs) with the combination of mRNA transfection to generate precise point mutations in human iPSCs. Using BEs, the authors successfully showed the single base alternation in hiPSCs with high efficiency, which was verified with fluorescent expression level and the Sanger sequencing. Furthermore, human disease-causing mutation in isogenic hiPSCs was generated with BEs, suggesting the possible experimental scheme to investigate the disease without genetic background effects. Lastly, genetic modification approach with PEs was introduced for hiPSCs, which allows more flexible and simultaneous multiple base alternation. The paper aimed to explore the feasibility of using BEs or PEs in hiPSCs, and the authors illustrated that such genetic modification strategy works well in hiPSCs compared to the conventional Cas9 system. However, more thorough examination into the usage of BEs or PEs in hiPSCs would have made the paper more resourceful.
Reply: We thank reviewer #2 for the constructive criticism, which we have addressed as shown below:
Specific comments
In this paper, the conventional CRISPR/Cas9 system was shown to perform less efficient than newly introduced systems in hiPSCs, yet the experimental data for the previous CRISPR/Cas9 system was weakly demonstrated. For example, one of the major improvements is introduction of in vitro transcribed BE mRNA instead of BE-expressing plasmid or protein. Also, the references and comparisons to the other papers would be very helpful.
Reply: We agree with the reviewer that a more detailed comparison to the conventional CRISPR/Cas9 nuclease system would improve the manuscript. To extend the comparison we have added a new Supplementary Table (now Table S2; page 27), which documents the editing rates of the SAMHD1 gene comparing plasmid vs. mRNA transfection.
The comparison of transfection efficiency among the different types of media for hiPSCs transfection implied that the culture condition should be carefully considered. Since the paper intended to focus on the usage of new genetic alternation technologies in hiPSCs, the comparison should be made more precisely. Additionally, the insights into why different types of media resulted in varying transfection efficiency would benefit the readers.
Reply: Unfortunately, we cannot provide further insight into why different types of media resulted in varying transfection efficiencies. The suppliers of the different media do not provide a full list of the ingredients, so it is not possible for us to determine which of the substances leads to the differences in transfection efficiencies. Nevertheless, we believe that even without this information, readers benefit from knowing that different hiPSC culture conditions should be considered when planning the experiment.
Statistical methods or p-values should be noted when statistical comparison was made (e.g. Figure 1D).
Reply (page 5, line44): We now describe the statistical methods in more detail in the revised manuscript, thank you.
Overall, the portion of fluorescently labeled cells was very high when using BEs for gene editing, indicating the highly efficient rate of transfection. In addition, the sequencing data also informed that number of edited cells is greater. In order to convey that the new technologies allow the greater transfection efficiency in hiPSCs, the quantitative results should be compared to the previous research, in which various types of gene editing technology were used for hiPSCs.
Reply: We have added three references of previous research (20-22), that had employed the conventional CRISPR/Cas9 nuclease approach in hiPSCs in the discussion.
Reviewer 3 Report
The study by Surun D. described the application of mRNA delivery-based CRISPR base editors and prime editors for site-specific editing in human iPSCs. Using traditional CRISPR HDR-based strategy, the study encountered no efficiency of correcting the point mutation in iPSCs tested. The authors then turned to test the mRNA-based ABE and CBE system for introducing base substitutions. Both ABE and CBE were firstly tested using HEK293T cells using the EGFP gene. With the successful demonstration in HEK cells, a further optimization of mRNA transfection was optimized for human iPSCs. This optimization is of great importance for the stem cells field and will be useful. The research team demonstrated the applicability of the mRNA-based ABE and CBE in hiPSCs using three independent experiments: EGFP-to-BFP conversion, SAMHD1 point mutation correction, and TP53 cancer-causing mutations introduction. All experiments proved that the mRNA delivery approach work very efficient in the hiPSCs. And finally, the research team also tested the recent Prime editing technology, which also worked quite efficient.
In general, this is a very well conducted study. Although the study might not possess very high technological and biological novelty, it provides a very well and systematic demonstration and guidelines for applying the base and prime editors in human iPSCs. The results presented by the study are of great interests for the stem cells and gene editing fields as well. Also, the study is very well written. I do not really feel there is a need for major changes.
Figure 1a and 2a, the 5’Cap cartoon, maybe consider to remove the shape. Simply just use the text.
Figure 6a, the gRNA (in black) looks strange, should be draw more nicely.
Author Response
Reviwer3:
Comments and Suggestions for Authors
The study by Surun D. described the application of mRNA delivery-based CRISPR base editors and prime editors for site-specific editing in human iPSCs. Using traditional CRISPR HDR-based strategy, the study encountered no efficiency of correcting the point mutation in iPSCs tested. The authors then turned to test the mRNA-based ABE and CBE system for introducing base substitutions. Both ABE and CBE were firstly tested using HEK293T cells using the EGFP gene. With the successful demonstration in HEK cells, a further optimization of mRNA transfection was optimized for human iPSCs. This optimization is of great importance for the stem cells field and will be useful. The research team demonstrated the applicability of the mRNA-based ABE and CBE in hiPSCs using three independent experiments: EGFP-to-BFP conversion, SAMHD1 point mutation correction, and TP53 cancer-causing mutations introduction. All experiments proved that the mRNA delivery approach work very efficient in the hiPSCs. And finally, the research team also tested the recent Prime editing technology, which also worked quite efficient. In general, this is a very well conducted study. Although the study might not possess very high technological and biological novelty, it provides a very well and systematic demonstration and guidelines for applying the base and prime editors in human iPSCs. The results presented by the study are of great interests for the stem cells and gene editing fields as well. Also, the study is very well written. I do not really feel there is a need for major changes.
Reply: We thank reviewer #3 for this concise assessment. We have addresses the suggestions as follows:
Figure 1a and 2a, the 5’Cap cartoon, maybe consider to remove the shape. Simply just use the text.
Reply (page 8 and 9): We have redrawn the illustration and removed the 5’Cap cartoon as suggested.
Figure 6a, the gRNA (in black) looks strange, should be draw more nicely.
Reply (page 16): We have improved the pegRNA drawing in Figure 6a, thank you.